# Better Algorithms for Individually Fair $k$-Clustering

**Deeparnab Chakrabarty**[*]
Department of Computer Science
Dartmouth College
Hanover, NH 03755
deepc@cs.dartmouth.edu

**Maryam Negahbani**
Department of Computer Science
Dartmouth College
Hanover, NH 03755
maryam@cs.dartmouth.edu

## Abstract

We study data clustering problems with $\ell_p$-norm objectives (e.g. $k$-MEDIAN and $k$-MEANS) in the context of individual fairness. The dataset consists of $n$ points, and we want to find $k$ centers such that (a) the objective is minimized, while (b) respecting the individual fairness constraint that every point $v$ has a center within a distance at most $r(v)$, where $r(v)$ is $v$'s distance to its $(n/k)$th nearest point. Jung, Kannan, and Lutz [FORC 2020] introduced this concept and designed a clustering algorithm with provable (approximate) fairness and objective guarantees for the $\ell_\infty$ or $k$-CENTER objective. Mahabadi and Vakilian [ICML 2020] revisited this problem to give a local-search algorithm for all $\ell_p$-norms. Empirically, their algorithms outperform Jung et. al.'s by a large margin in terms of cost (for $k$-MEDIAN and $k$-MEANS), but they incur a reasonable loss in fairness. In this paper, our main contribution is to use Linear Programming (LP) techniques to obtain better algorithms for this problem, both in theory and in practice. We prove that by modifying known LP rounding techniques, one gets a worst-case guarantee on the objective which is much better than in MV20, and empirically, this objective is extremely close to the optimal. Furthermore, our theoretical fairness guarantees are comparable with MV20 in theory, and empirically, we obtain noticeably fairer solutions. Although solving the LP *exactly* might be prohibitive, we demonstrate that in practice, a simple sparsification technique drastically improves the run-time of our algorithm.

## 1 Introduction

As machine learning algorithms are widely used in practice for making high-stakes decisions affecting human lives, there has been a huge body of work on FAIR-ML trying to ensure 'fairness' in the solutions returned by these algorithms. There are two large intersecting bodies of work : one body's main focus in to understand what 'fairness' means (e.g. [26, 39, 38, 36, 14, 25]) in various different contexts, and the second body's focus has been on addressing the *algorithmic challenges* brought forth by these considerations (e.g., [24, 9, 33, 6, 29, 5]).

This paper falls in the second class. In particular, we consider an *individual fairness* model proposed by Jung, Kannan, and Lutz [25] for a $k$-clustering problem. Given points (clients) $X$ in a space with metric distance $d$, find $k$ points (facilities) $S \subseteq X$, minimizing $\left( \sum_{v \in X} d(v, S)^p \right)^{1/p}$ where $d(v, S)$ is $v$'s distance to the closest point in $S$. This includes $k$-CENTER, $k$-MEDIAN and the popular $k$-MEANS objective for $p = \infty$, $p = 1$ and $p = 2$ respectively, problems which have been extensively studied [12, 23, 27, 4, 2] in the algorithms literature. Jung et. al. [25] proposed that in this context a solution would be deemed individually fair, if for every client $v \in X$ there is an open facility not too far from it. More precisely, if there is a facility within distance $r(v)$ which is the smallest radius

---

[*]Supported by NSF grant #2041920

around $v$ that contains $n/k$ points. The rationale behind $n/k$ is that if all the points had equal chance of getting picked as a facility, each client $v$ would expect a facility to be open within distance $r(v)$. Furthermore, this formulation encourages a solution to open more facilities in dense areas of the metric which is motivated by the facility location application of clustering.

Jung et al. [25] gave a solution where every client $v$ was served within a radius of $2r(v)$, which as a jargon is called 2-approximate fair solution. However, their solution did not explicitly consider the "objective" function ($k$-MEANS/$k$-MEDIAN, for instance) in the clustering problem, which is often used as a proxy to measure the quality of the clustering. This was addressed in a follow up paper by Mahabadi and Vakilian [29] who gave a $(7, O(p))$-approximation with respect to the $\ell_p$-norm objective. That is, they give a local-search based solution which is 7-approximately fair, but the objective is violated by some $Cp$-factor where the constant $C$ is rather large (for $p = 1$, the factor is 84). The theoretical running time of their algorithm is $\tilde{O}(pk^5n^4)$.

## 1.1 Our Contributions

The main contribution of our paper is to give improved algorithms for this problem using *linear programming rounding*. Our study stems from two observations: one, that the problem at the core of Jung et al. [25] was in fact studied as "weighted/priority $k$-CENTER problem" by Plesník[32], and that if all the $r(v)$'s were the same (which may not at all be the case), then the clustering problem has also been studied under the guise of centridian/ordered median problem [3, 8, 10]. Combining ideas from these two bodies of work, we design an $(8, 8)$ algorithm for the FAIR-$k$-MEDIAN problem, which obtains an 8-approximation for both cost and fairness (our cost guarantees improve as $p$ grows).

**Result 1.** There is an $(8, 2^{1+2/p})$-approximation algorithm for FAIR-$(p, k)$-CLUSTERING that runs in LP solving time plus $\tilde{O}(n^2)$, overall $\tilde{O}(kn^4)$. In particular, we have $(8, 8)$-approximation and $(8, 4)$-approximation algorithms for FAIR-$k$-MEDIAN and FAIR-$k$-MEANS, respectively.

**Remark.** The approximation factor compares itself with the optimal solution which opens centers at the points in $X$. This is also the notion considered by [29] and [25]. If the optimal solution is allowed to open centers anywhere in the real space, then one has an additional factor 2 loss in the approximation [15]. Whether this extra hit of factor 2 is necessary, we leave as an interesting algorithmic question.

Although solving an LP may seem prohibitive in practice, we can obtain a much faster running time by implementing a *sparsification* routine (inspired by [32, 21]) with a marginal hit in the fairness and clustering cost (see Lemma 5 for details). Empirically, this greatly decreases the running time, and is often faster than the [29] implementation.

In our experiments, we also find that our theoretical bounds are too pessimistic. Indeed, we show that our algorithm's cost is at most %1 more than the *optimal* clustering cost (which does not have any fairness violation), almost always, and never more than %15 in the rest. Furthermore, our maximum fairness violation is at most a factor of 1.27 which is much better than our theoretical guarantee of 8. We also do a more fine-grained analysis of the fairness violation : consider a vector where each coordinate stands for a clients "unfairness" indicating the ratio of its distance to $r(v)$. When we plot this as a histogram, we find that most of the mass is shifted to the "left", that is, the percentage of clients who satisfy their fairness constraints is significantly larger than in the [29] solution. This seems to suggest the linear program, which is trying to minimize the cost, itself tries to increase the number of fairly treated clients. We leave a theoretical investigation of this phenomenon for future work.

Our experiments also demonstrate the *price of fairness*. We find that our linear programs, which maintains absolute fairness, have objective value considerably larger than that of [29]. On the other hand, if we tune the "fairness violation" of the linear program to match that of [29], then the objective value of our algorithm drops. We run experiments to further elaborate on the inherent *cost of fairness* in our datasets by demonstrating how the optimal cost changes with respect to varying degrees of fairness relaxation.

It is worth noting that our algorithm works for arbitrary values of $r(v) \geq 0$ for points $v$ (and this may be true for [29] as well), and we present our results thus. This setting might be of interest in applications where the "fair radius" may not be $n/k$ but something more nuanced.

## 1.2 Other related work

$k$-CENTER has a 2-approximation due to Gonzales, and Hochbaum and Shmoys [17, 21] and they prove it is NP-hard to get better approximations. $k$-MEDIAN, and $k$-MEANS are hard to approximate within factors better than 1.73 and 3.94 [19] respectively with current best approximations being 2.67 by [7] and 9 by [2]. Also recently, there has been an improvement on lower-bounds for approximating *Continuous* $k$-MEDIAN and $k$-MEANS where centers can be picked anywhere in the real space. By Cohen-Addad, Karthik, and Lee [15], it is NP-hard to approximate Continuous $k$-MEDIAN and $k$-MEANS within factors $2 - o(1)$ and $4 - o(1)$ respectively.

FAIR-$k$-CENTER is a special case of *Priority* $k$-CENTER where the radii $r(v)$ in which a point $v$ demands a center at that distance, are general values. [32] introduced this problem and gave a best possible 2-approximation. Gørtz and Wirth [18] study the problem for asymmetric metrics and prove that it is NP-hard to obtain any non-trivial approximation. [5] give a 9-approximation for the problem in presence of *outliers* and further generalize to constant approximations for general constraints on the solution centers. Another very closely related problem is *Chance-k-Coverage* introduced in [20] in which for any point $v$, in addition to $r(v)$, a probability $p(v)$ is given and the goal is to find a distribution on possible solutions such that a solution drawn from this distribution covers $v$ with probability at least $p(v)$. This also has a 9-approximation by [20].

A clustering problem related to FAIR-$k$-MEDIAN and FAIR-$k$-MEANS is the *Simultaneous k-Clustering* in which the goal is to find a solution with approximation guarantees with respect to any monotone, symmetric norm. This problem has an $O(1)$-approximation due to Chakrabarty and Swamy [11] with a line of previous work including [3, 10, 8].

Another similar notion of individual fairness is introduced by Chen et al. [13] in which a solution is fair if there is no group of size at least $n/k$ for which there exists a facility that would reduce the connection cost of all members of the group if opened. [13] give a $(1 + \sqrt{2})$-approximation for $\ell_1$, $\ell_2$, and $\ell_\infty$ norm distances for the setting where facilities can be places anywhere in the real space. Micha and Shah [30] modified the approach to give close to 2-approximation for $\ell_2$ and proved the previous results for $\ell_1$ and $\ell_\infty$ are indeed tight.

Recently, Abbasi et al. [1] introduced a new notion of fairness centered around providing equitable group representations. In this problem, the points are partitioned into groups and the goal is to minimize the maximum clustering cost a group can have in the solution. The idea is that the cluster centers should represent each group fairly, and this is captured by ensuring that each group's clustering cost is comparable with the others. This notion is orthogonal to the fairness criteria we study in this paper; Although we could tune the maximum distance the points in a group can have to our solution, that does not give any guarantees with respect to the objective cost and vice versa.

## 2 Preliminaries

In this section, we formally define our problems, establish some notations, and describe a classic clustering routine due to Hochbaum and Shmoys [21] with modifications by Plesník[32]. Given a subset $S \subseteq X$, we use $d(v, S)$ to denote $v$'s distance to the closest point in $S$.

**Definition 1** (FAIR-$(p, k)$-CLUSTERING Problem). The input is a metric space $(X, d)$, radius function $r : X \to \mathbb{R}^+$, and integers $p, k \geq 1$. The goal is to find $S \subseteq X$ of size at most $k$ such that $d(v, S) \leq r(v)$ for all $v \in X$ and the clustering cost $\left( \sum_{v \in X} d(v, S)^p \right)^{1/p}$ is minimized.

Let opt be the clustering cost of an optimal solution. For $\alpha, \beta \geq 1$, an $(\alpha, \beta)$-approximate solution is $S \subseteq X$ of size at most $k$ with $d(v, S) \leq \alpha r(v)$ for all $v \in X$ and $\left( \sum_{v \in X} d(v, S)^p \right)^{1/p} \leq \beta$opt. In plain English, the fairness approximation is $\alpha$ while the objective/cost approximation is $\beta$. Our main result is the following.

**Theorem 1.** *There is an $(8, 2^{1+2/p})$-approximation algorithm for* FAIR-$(p, k)$-CLUSTERING *that runs in LP solving time plus $\tilde{O}(n^2)$, overall $\tilde{O}(kn^4)$.*

The algorithm relies on rounding a solution to the following LP for FAIR-$(p, k)$-CLUSTERING[2] where the optimal LP objective is at most opt$^p$. The variable $y_u$ denotes the amount by which $u \in X$

---

[2]The LP and its rounding is slightly different for $p = \infty$; we omit this from this version.

is open as a center. $x_{vu}$ for $v, u \in X$ is the amount by which $v$ is assigned to $u$. The constraints respectively capture the conditions: every client must connect to someone, $k$ centers are opened, no client can connect to an unopened center, and crucially that a client cannot travel to a center further than $r(v)$.

$$\mathsf{opt}^p \geq \min \sum_{v,u \in X} d(v,u)^p x_{vu} \qquad \text{(LP)}$$

$$\sum_{u \in X} x_{vu} = 1, \ \forall v \in X \qquad \text{(LP1)}$$

$$\sum_{u \in X} y_u = k \qquad \text{(LP2)}$$

$$x_{vu} \leq y_u, \qquad \forall v, u \in X \qquad \text{(LP3)}$$

$$x_{vu} = 0, \forall v, u : d(v,u) > r(v) \qquad \text{(LP4)}$$

$$0 \leq x_{vu}, y_u \leq 1, \qquad \forall v, u \in X.$$

From here on, we use the notation $B(u, r)$ for $u \in X$ and $r \in \mathbb{R}^+$ to denote the points in a ball of radius $r$ around $u$. That is $B(u, r) := \{v \in X : d(u,v) \leq r\}$. Also, for any set of points $U \subseteq X$, let $y(U) := \sum_{u \in U} y_u$. Considering (LP4) and (LP1) the following holds.

**Fact 1.** If $y$ is from a feasible LP solution, then $y(B(v, r(v))) \geq 1$ for all $v \in X$.

We now describe a routine Filter due to [32, 21] which is used as a subroutine in our main algorithm. Assume all the points are initially "uncovered". The routine takes a function $R : X \to \mathbb{R}^+$, sorts the points in order of *increasing* $R(v)$'s. Then it considers the first point in this order, calls it a "representative" and "covers" all the points $v$ at distance at most $2r(v)$ from it. Call these points $D(v)$. Repeating this procedure until all the points are covered, forms a partition on $X$ such that the representatives are "far apart", while each non-representative is assigned to a "nearby" representative, among other useful properties listed in Fact 2 and Fact 3.

---

**Algorithm 1** Filter

**Input:** Metric $(X, d)$, radius function $R : X \to \mathbb{R}^+$
1: $U \leftarrow X$             $\triangleright$ The set of "uncovered points"
2: $S \leftarrow \emptyset$            $\triangleright$ The set of "representatives"
3: **while** $U \neq \emptyset$ **do**
4:      $u \leftarrow \arg\min_{v \in U} R(v)$      $\triangleright$ The first point in $U$ in non-decreasing $R$ order
5:      $S \leftarrow S \cup u$
6:      $D(u) \leftarrow \{v \in U : d(u,v) \leq 2R(v)\}$    $\triangleright$ $D(u) = B(u, 2R(v)) \cap U$ and includes $u$ itself
7:      $U \leftarrow U \backslash D(u)$
8: **end while**
**Output:** $S, \{D(u) : u \in S\}$

---

**Fact 2.** [32, 21] The following is true for the output of Filter: (a) $\forall u, v \in S, d(u,v) > 2\max\{R(u), R(v)\}$, (b) The balls $\{B(u, R(u)) : u \in S\}$ are mutually disjoint, (c) The set $\{D(u) : u \in S\}$ partitions $X$, (d) $\forall u \in S, \forall v \in D(u), R(u) \leq R(v)$, and (e) $\forall u \in S, \forall v \in D(u), d(u,v) \leq 2R(v)$.

**Fact 3.** For any $u \in S$ and $w \in B(u, R(u))$, the unique closest point in $S$ to $w$ is $u$.

*Proof.* Suppose otherwise. That is, there exists $v \in S$ not equal to $u$ s.t. $d(w, v) \leq d(w, u)$. Then $d(u, v) \leq d(u, w) + d(w, v) \leq 2d(w, u) \leq 2R(u)$ which contradicts Fact 2 as we must have $d(u, v) > 2\max\{R(u), R(v)\}$. $\qquad \square$

To elaborate on the importance of Algorithm 1 and build some intuition, we point out the following theorem of [32, 25].

**Theorem 2.** *Take $S$ the output of Algorithm 1 on a* FAIR-$(p, k)$-CLUSTERING *instance with $R := r$. Then if the instance is feasible, $|S| \leq k$ and $d(v, S) \leq 2r(v)$ for all $v \in X$.*

*Proof.* By Fact 2 we know that $\{D(u) : u \in S\}$ partitions $X$ so for any $v \in X$, there exists a $u \in S$ for which $v \in D(u)$. Plus, $d(v, u) \leq 2R(v) = 2r(v)$. Now it only remains to prove $|S| \leq k$. To see this, let $S^*$ be some feasible solution and observe that two different $u, w \in S$, cannot be

covered by *the same* center in $S^*$. Since otherwise, if there exists $f \in S^*$ for which $d(u, f) \leq r(u)$ and $d(w, f) \leq r(w)$ by triangle inequality $d(u, v) \leq d(u, f) + d(f, w) \leq r(u) + r(v)$ and this contradicts $d(u, w) > 2 \max\{r(u), r(v)\}$ from Fact 2. $\qquad\square$

But of course, the above theorem does not give any guarantees for the clustering cost (unless $p = \infty$). It might be the case that many points are paying close to 0 towards the clustering cost in the optimal solution, but are made to connect to a point much farther in the above procedure.

## 3 Algorithm for FAIR-$(p, k)$-CLUSTERING problem

Now we are ready to describe our algorithm Fair-Round which establishes Theorem 1. At a high-level, we run our Filter routine by defining the input function $R$ in a manner that is conscious of the FAIR-$(p, k)$-CLUSTERING cost: given $(x^*, y^*)$ which is an optimal solution to (LP), for any $v \in X$ let $C_v$ be $v$'s contribution to the LP cost i.e. $C_v := \sum_{u \in X} d(v, u)^p x^*_{vu}$ and define $R(v) := \min\{r(v), (2C_v)^{1/p}\}$. Let us ponder for a bit to see what changes from Theorem 2. For the output $S$, we still have the fairness guarantee $d(v, S) \leq 2r(v)$ for all $v$ but since $(2C_v)^{1/p}$ might be less than $r(v)$ for any $v$, we cannot guarantee that $|S| \leq k$. Thankfully, in this case, we can still prove $|S| \leq 2k$ (Corollary 1). The rest of the algorithm is deciding on a subset of at most $k$ points out of this $S$ to return as the final solution, while ensuring the fairness and cost guarantees are still within constant factor of the optimal. This idea is very similar to existing ideas in [3, 12] which look at the problem without fairness considerations.

Recall that $\{D(u) : u \in S\}$ partitions $X$ and each $u \in S$ is responsible for covering all the points in $D(u)$. Here, we could simply move each point in $D(u)$ to $u$ and divert the $y$ value of each point to its closest point in $S$. Note that, $y(S) = k$ (Fact 4) and similar to the proof of Theorem 2 we could show $y_u \geq 1/2$ (Lemma 1) for all $u \in S$ hence $|S| \leq 2k$ (Corollary 1). If this leads to some $y$-value reaching 1, we open those centers.

For $u \in S$ with $y_u < 1$, if we do not decide to include it in the final solution, we promise to open $S_u$, its closest point in $S$ other than itself. In this case, all the $|D(u)|$ points on $u$ are delegated to $S_u$. Using the fact that $y_u < 1$, we can prove that fairness guarantee (Lemma 3) approximately holds for the points in $D(u)$ even after this delegation.

To get the clustering cost guarantee, we need to do more work. Observe that, currently, $u$ is already fractionally assigned to $v$ by $1 - y_u$. So if instead of $y_u \geq 1/2$ we had $y_u = 1/2$ we could ensure that already $u$ is assigned to $v$ by $1/2$ thus integrally assigning $u$ to $v$ only doubles the cost. This is why we need to do more work to get $y_u \in \{1/2, 1\}$ for $u \in S$ (see Lemma 2) and then bound the clustering cost in Lemma 4.

**Fact 4.** $y(S) = k$ and remains so after Line 11.

*Proof.* We initialize $y$ to $y^*$ for which $y^*(X) = k$ by LP2. The algorithm then transfers $y(X)$ to $S$ by Line 11 without changing the total. Afterwards, the modifications in Lines 13 and 19 move the $y$ mass around in $S$ but the total value $y(S)$ remains unchanged. $\qquad\square$

**Lemma 1.** After Line 14 of Algorithm 2, $1/2 \leq y_u \leq 1$ for all $u \in S$.

*Proof.* First we argue that $y_u \geq 1/2$ for all $u \in S$ by the end of Line 11. Fix $u \in S$. Per Fact 3 $y(B(u, R(u)))$ is entirely moved to $y_u$. By definition of $R(u)$ there are two cases: Case I, $R(u) = r(u)$ thus $y_u \geq 1$ as $y^*(B(v, r(v))) \geq 1$ for all $v \in X$ according to Fact 1. Case II, $R(u) = (2C_u)^{1/p}$ then by Markov's inequality $y^*(B(u, R(u))) \geq 1/2$ and after this point, $y_u$ is never decreased to below 1/2. More precisely, $C_u = \sum_{v \in X} d(u, v)^p x^*_{uv} \geq \sum_{\substack{v \in X: \\ d(u,v) > R(u)}} d(u, v)^p x^*_{uv} \geq 2C_u \sum_{\substack{v \in X: \\ d(u,v) > R(u)}} x^*_{uv}$. Considering $\sum_{v \in X} x^*_{uv} = 1$ by LP1, this implies $\sum_{\substack{v \in X: \\ d(u,v) \leq R(u)}} x^*_{uv} \geq 1/2$ thus $y^*(B(u, R(u))) \geq 1/2$.

As for proving $y_u \leq 1$, it might indeed be the case that $y_u > 1$ by the end of Line 11 but the loop ending at Line 14 can guarantee $y_u \leq 1$ for all $u \in S$. This is because $y(S) = k$ (Fact 4) and we already checked in the beginning of Algorithm 2 that $|S| > k$. $\qquad\square$

**Corollary 1.** $S$ in Algorithm 2 has size at most $2k$.

---

**Algorithm 2** Fair-Round: FAIR-$(p, k)$-CLUSTERING bi-criteria approximation

---

**Input:** Metric $(X, d)$, radius function $r : X \to \mathbb{R}^+$, and $(x^*, y^*)$ an optimal solution of LP

  1: $C_v \leftarrow \sum_{u \in X} d(v, u)^p x_{vu}^* \quad \forall v \in X$                             $\triangleright$ $v$'s cost share in the LP objective

  2: $R(v) \leftarrow \min\{r(v), (2C_v)^{1/p}\} \quad \forall v \in X$

  3: $S, \{D(u) : u \in S\} \leftarrow \mathsf{Filter}((X, d), R)$

  4: **if** $|S| \le k$ **then**

  5:      **return** $S$

  6: **end if**

  7: $(x, y) \leftarrow (x^*, y^*)$

  8: **for all** $v \in X \backslash S$ **do**                $\triangleright$ Direct $y$ mass from outside of $S$ to the closest point in $S$

  9:      $u \leftarrow$ closest point in $S$ to $v$

10:      $y_u \leftarrow y_u + y_v, y_v \leftarrow 0$              $\triangleright$ Note: May cause $y_u$ to increase above 1

11: **end for**

12: **while** There are $u, v \in S$ with $y_u > 1$ and $y_v < 1$ **do**        $\triangleright$ Ensure $y_u \le 1$ for all $u \in S$

13:      $y_u \leftarrow y_u - \delta, y_v \leftarrow y_v + \delta$, where $\delta := \min\{1 - y_v, y_u - 1\}$

14: **end while**          $\triangleright$ Remark: By now, $1/2 \le y_u \le 1$ for all $u \in S$ (see Lemma 1).

15: $S_u \leftarrow$ closest point in $S \backslash u$ to $u \quad \forall u \in S$

16: **while** There are $u, v \in S$ with

17:      $1/2 < y_u < 1$, and $y_v < 1$,

18:      $d(u, S_u)^p |D(u)| > d(v, S_v)^p |D(v)|$ **do**       $\triangleright$ Move $y$ mass from $u$ to $v$ if $u$ is costlier

19:      $y_u \leftarrow y_u - \delta, y_v \leftarrow y_v + \delta$, where $\delta := \min\{1 - y_v, y_u - 1/2\}$

20: **end while**        $\triangleright$ Remark: At this point, $y_u \in \{1/2, 1\}$ for all $u \in S$ (see Lemma 2).

21: $T \leftarrow u \in S$ with $y_u = 1$

22: Consider the forest of arbitrary rooted trees on vertices $u \in S$ with edges $(u, S_u)$. Let $O$ be odd-level vertices in $S \backslash T$, and $E$ be even-level vertices in $S \backslash T$.

23: **if** $|E| \le |O|$ **then**

24:      $T \leftarrow T \cup E$

25: **else**

26:      $T \leftarrow T \cup O$

27: **end if**

**Output:** $T$

---

**Lemma 2.** After Line 20 of Algorithm 2, $y_u \in \{1/2, 1\}$ for all $u \in S$.

*Proof.* Suppose not. Since $y(S) = k$ (Fact 4) and $k$ is an integer, by Lemma 1, there has to be at least two $u, v \in S$ with $y_u, y_v \in (1/2, 1)$ which is a contradiction, since either one of them has to be changed to $1/2$ or $1$ in the while loop ending at Line 20. $\square$

**Lemma 3.** For all $v \in X$, $d(v, T) \le 8r(v)$.

*Proof.* Fix $v \in X$. Since $\{D(u) : u \in S\}$ partitions $X$ there exists $u \in S$ such that $v \in D(u)$. According to Fact 2 $d(v, u) \le 2R(v) \le 2r(v)$ by definition of $R$. If $u$ ends up in $T$ we are done. Else, it has to be that $y_u < 1$ and $S_u \in T$. In what follows, we prove that if $y_u < 1$ at Line 20 then $d(u, S_u) \le 6r(v)$. This implies $d(v, S_u) \le 8r(v)$ hence the lemma.

We know that initially $y^*(B(v, r(v))) \ge 1$ per Fact 1 but since $v \notin S$, the $y$ mass on $B(v, r(v))$ has been moved to $S$ by Line 11. If for all $w \in B(v, r(v))$ their closest point in $S$ was $u$, all of $y(B(v, r(v)))$ would be moved to $u$ then $y_u = 1$ by the end of Line 20. So there must exist $w \in B(v, r(v))$ with $x_{vw}^* > 0$ along with $u' \in S, u' \neq u$, such that $d(w, u') \le d(w, u)$ which made $y_w$ to be moved to $u'$. By definition of $S_u$, $d(u, S_u) \le d(u, u')$. Applying the triangle inequality twice gives:

$$
\begin{aligned}
d(u, S_u) &\le d(u, u') \le d(u, w) + d(w, u') \le 2d(u, w) \\
&\le 2(d(u, v) + d(v, w)) &\le 2(2r(v) + r(v)) = 6r(v)
\end{aligned}
$$

where the last inequality comes from Fact 2 stating $d(u, v) \le 2r(v)$, and from the fact that $x_{vw}^* > 0$ implies $d(v, w) \le r(v)$ by LP4. $\square$

**Lemma 4.** $\left( \sum_{v \in X} d(v, T)^p \right)^{1/p} \le \left( (2^{p+2}) \sum_{v, u \in X} C_v \right)^{1/p} \le 2^{1+2/p}\mathsf{opt}$.

*Proof.* For the proof, we compare $\sum_{v\in X} d(v,T)^p$ with the optimal LP cost $\sum_{v\in X} C_v$ which is at most $\mathsf{opt}^p$. Fix $v \in X$ and $u \in S$ for which $v \in D(u)$. By Fact 2 $d(v,u) \leq 2R(v) \leq 2(2C_v)^{1/p}$ per definition of $R$. So moving $D(u)$ to $u$ for all $u \in S$ has an additive cost of $\sum_{u\in S}\sum_{v\in D(u)} d(v,u)^p \leq 2^{p+1}\sum_{v\in X} C_v$. From now on, assume there are $D(u)$ collocated points on a $u \in S$.

Moving around the $y$ mass up to Line 11 adds a multiplicative factor of $2^p$ loss in the approximation ratio. The logic is: if $u \in S$ was relying on $w \in X$ in the LP solution, meaning, $x^*_{uw} > 0$ and $y_w$ was moved to a $u' \in S$ (due to $d(w,u') \leq d(w,u)$) then the cost $u$ has to pay to connect to $u'$ is $d(u,u')^p \leq (d(u,w) + d(w,u'))^p \leq 2^p d(u,w)^p$ which is a $2^p$ factor worse than the LP cost $d(u,w)^p$ it was paying to connect to $w$ earlier.

At this point (Line 14), note that the cost incurred by $u$ is $|D(u)|d(u,S_u)^p(1 - y_u)$. To elaborate on this, corresponding to $y$, we define an $x$ such that $(x,y)$ is feasible for LP. Let $x_{uu} = y_u$ and $x_{uS_u} = 1 - y_u$ so that LP4 is satisfied for $u$. Then the cost incurred by u is $|D(u)|(d(u,u)^p x_{uu} + d(u,S_u)^p x_{uS_u}) = |D(u)|d(u,S_u)^p(1 - y_u)$. The while loop ending at Line 20 does not increase the value of the objective function: As we decrease $y_u$ and increase $y_v$, only if the cost incurred by $u$ is bigger than that of $v$. The last multiplicative factor 2 loss comes from when $u \in S\backslash T$. In which case, $y_u = 1/2$ and $S_u \in T$. So by assigning $u$ to $S_u$ we pay $|D(u)|d(u,S_u)^p$ which is twice more than before (as $1 - y_u = 1/2$). Observe that this last step is why we needed to do all the work to get $y_u \in \{1/2, 1\}$ in Lemma 2. Putting the three steps together, the overall cost is at most $2 \times 2^p + 2^{p+1} = 2^{p+2}$ times the LP cost or at most $2^{1+2/p}\mathsf{opt}$. $\qquad\square$

*Proof of Theorem 1.* Using Corollary 1 and the fact that $|T| \leq |S|/2$ by construction, we have $|T| \leq k$. Lemmas 3 and 4 give the fairness and approximation guarantees. As for runtime, notice that Algorithm 2 runs in time $\tilde{O}(n^2)$. But the runtime is dominated by the LP solving time. According to [37], finding a $(1+\varepsilon)$-approximation to LP takes time $O(kn^2/\epsilon^2)$. Setting $\varepsilon = 1/n$ gives the $O(kn^4)$ runtime. $\qquad\square$

As evident, the runtime is dominated by the LP solving time. We end this section by desciding the sparsification pre-processing that when applied to the original instance, can tremendously decrease the LP solving time in practice while incurring only a small loss in fairness and clustering cost. One reason why the LP takes a lot of time is because there are many variables; $x_{vu}$ for every $v \in X$ and every $u$ within distance $r(v)$ of $v$. To fix this, we first run the Filter algorithm on the data set with $R(v) = \delta r(v)$, where $\delta$ is a tune-able parameter. It is not too hard to quantify the loss in fairness and cost as a function of $\delta$, and we do so in the lemma below. More importantly, note that when $\delta = 1$, then the number of variables goes down from $n$ to $O(k)$; this is because of the definition of $r(v)$ which guarantees $\approx n/k$ points in the radius $r(v)$ around $v$. Therefore, running the pre-processing step would make the number of remaining points $\approx k$. In our experiments we set $\delta$ to be much smaller, and yet observe a great drop in our running times.

---

**Algorithm 3** Sparsification + Fair-Round

---

**Input:** Metric $(X,d)$, radius function $r : X \to \mathbb{R}^+$, parameter $\delta > 0$
 1: $S, \{D(u) : u \in S\} \leftarrow$ Filter$((X,d), \delta r)$
 2: $(x', y') \leftarrow$ solve LP only on points $S$ with objective function $\sum_{v,u\in S} d(v,u)^p x_{vu}|D(v)|$
 3: $x^*_{vw} \leftarrow x'_{uw} \forall v, w \in X, u \in S : v \in D(u)$ $\quad \triangleright$ $v$'s assignment is identical to its representative $u$
 4: $y^*_u \leftarrow y'_u$ if $u \in S$ and 0 otherwise
**Output:** Fair-Round$((X,d), (1+\delta)r, (x^*, y^*))$

---

**Lemma 5.** Algorithm 3 outputs an $(8(1+\delta), 2^{1+2/p}(1 + (\delta\phi)^p)^{1/p})$-approximation where $\phi = (\sum_{v\in X} r(v)^p)^{1/p}/\mathsf{opt}$.

*Proof.* Observe that $(x^*, y^*)$ is not a feasible LP solution anymore. Nevertheless, it will be feasible for when $r$ is dilated by a factor of $(1+\delta)$ in the LP4. Here is how we argue LP4 holds in this case: For any $v, w \in X$ for which $x^*_{vw} > 0$, recall $u \in S$ is chosen so $v \in D(u)$ and $x^*_{vw} := x'_{uw} > 0$ meaning $d(u,w) \leq r(u)$. We have $d(v,w) \leq d(v,u) + d(u,w) \leq \delta r(v) + r(u) \leq (1+\delta)r(v)$. The last two inequalities are by Fact 2 as $d(v,u) \leq \delta r(v)$ and $r(u) \leq r(v)$. As for the LP cost of $(x^*, y^*)$,

it is not longer upper-bounded by $\mathsf{opt}^p$ but rather by $\mathsf{opt}^p + \sum_{u \in S} \sum_{v \in D(u)} d(v, u)^p$. This additive term is at most $\sum_{v \in X} (\delta r(v))^p \leq (\delta \phi \mathsf{opt})^p$. Plugging this into Lemma 4 finishes the proof. $\qquad \square$

## 4 Experiments

**Summary.**  We run experiments to show that the empirical performance of our algorithms can be much superior to the worst-case guarantees claimed by our theorems (the codes are publicly availably on Github[3]). While implementing our algorithm, we make the following optimization : instead of choosing the constant "2" in Line 2 (definition of $R$) in our algorithm, we perform a binary-search to determine a better constant. Specifically, we find the smallest $\beta$ for which Algorithm 1 gives at most $k$ centers with $R(v) := \min\{r(v), (\beta C_v)^{1/p}\}$ for all $v$. This step is motivated by the experiments in [25].

We assess the performance with respect to (a) fairness, (b) objective value, and (c) running times. Due to space restrictions, we focus on the $k$-MEANS objective and leave $k$-MEDIAN results to Appendix B. Our fairness violation seldom exceeds 1.3 (compare this to the theoretical bound of 8), and often is much better than that of [29] and [25]. The objective value of our solution is in fact extremely close to the LP solution, which is a *lower bound* on the optimum cost (compare this to the theoretical bound of 4). This occurs because typically the LP solution itself has many integer entries. Finally, although the "vanilla" running time of the LP is pretty large, our sparsification routine Lemma 5 tremendously reduces the running time.

**Datasets.**  Similar to [29], we use 3 datasets from the UCI repository[4] [16] (also used in previous fair clustsering works of [14, 6]). We use a set of numeric features (matching previous work) to represent data in Euclidean metric.
**1-** `bank` [35] with 4,521 points, on data collected by the marketing campaign of a Portuguese banking institution. We use 3 features "age", "balance", and "duration".
**2-** `census` [28] with 32,561 points, based on the 1994 US census. Here we use the following 5 features: "age" , "final-weight", "education-num", "capital-gain", and "hours-per-week".
**3-** `diabetes` [34] with 101,766 points, from diabetes patient records from 130 hospitals across US.

**Benchmarks.**  We compare with the algorithms of [29] and [25]. For [29], we set parameters according to the experimental section of their paper. We remark that in their experimental section they perform 1 swap in their local search instead of the 4-swaps for which they prove their theorem. For our running time comparison, we also compare ours and the above two algorithm's running times with the popular $k$-Means$++$ algorithm of [4] from scikit learn toolkit [31] which has no fairness guarantees.

### 4.1  Fairness analysis

For any solution $T \subseteq X$, define the "violation array" $\theta_T$ over $v \in X$ as $\theta_T(v) = d(v, T)/r(v)$. This provides a more fine-grained view of the per-client fairness guarantee violation. Similar to [29] we use maximum violation (i.e. $\max_{v \in X} \theta_T(v)$) as a benchmark for fairness. In Figure 1 we have plotted the maximum violation of our algorithm Fair-Round, [29] (noted as MV), and [25] (noted as JKL). We find that our results are noticeably fairer than [29] while improving over [25] in many cases.

Next, we compare the histograms of the violation vectors $\theta$ as described above. This picture provides a better view of the fairness profile, as just looking at the maximum violation may not be a robust measure; an algorithm that is largely unfair may be preferred to an algorithm that is extremely unfair on a single point but extremely fair on all the rest. We show the histograms of the violation vectors of these algorithms in Figure 2 (for a full set of histograms see Appendix A.1). As an example, even though in `bank` with $k = 20$ our maximum violation is slightly worse than [25] our histogram shows that we are slightly fairer overall. We observe that our algorithm ensures complete fairness for at least %80 of the points in almost all the experiments. A theoretical study of the violation vector is interesting and left as future work.

---

[3]`https://github.com/moonin12/individually-fair-k-clustering`
[4]https://archive.ics.uci.edu/ml/datasets/

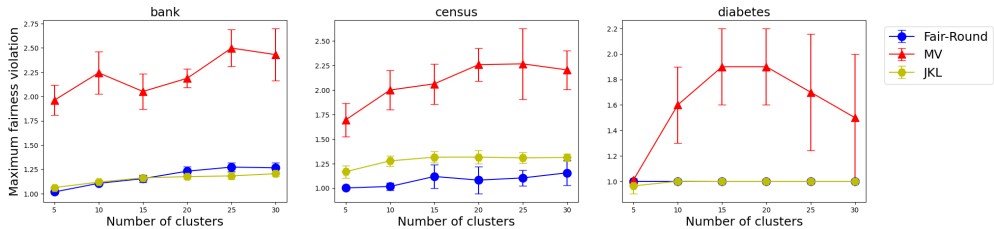

Figure 1: Comparison of maximum fairness violation with $k$-MEANS objective, between our algorithm Fair-Round, the algorithm in [25] (denoted as JKL), and the algorithm in [29] (denoted as MV), on average of 10 random samples of size 1000 each.

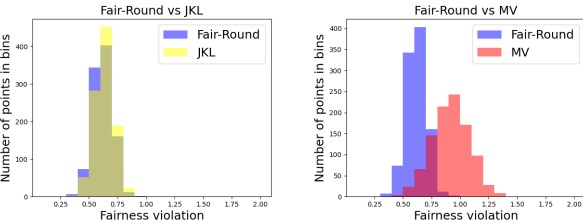

Figure 2: Histograms of violation vectors on bank with $k = 20$ $k$-MEANS objective comparing Fair-Round with the algorithm in [25] (denoted as JKL) and the algorithm in [29] (denoted as MV), on average of 10 random samples of size 1000 each.

## 4.2 Cost analysis

In Figure 3 we demonstrate that in our experiments, the objective value almost matches the optimal cost, begin at most %1 more almost always, and never above %15 more than the LP cost. Note that our cost is also *higher* than that of [29], and the reader may be wondering how the latter can be better than the optimum cost. The reason is that the [29] cost is violating the fairness constraint while the optimal cost is not. To do a more apples-to-apples comparison, one can also allow the same violations as [29] and re-solve the linear program and also our rounding algorithms. On doing so, we do find that our algorithm's cost becomes lower than that of [29]. Details of this can be found in Appendix A.2. The set-up also allows us to measure the *cost of fairness*: how much does the linear programming cost and our algorithm's cost decrease as we relax the fairness constraints. We do this empirical study; see Appendix A.3 for our results.

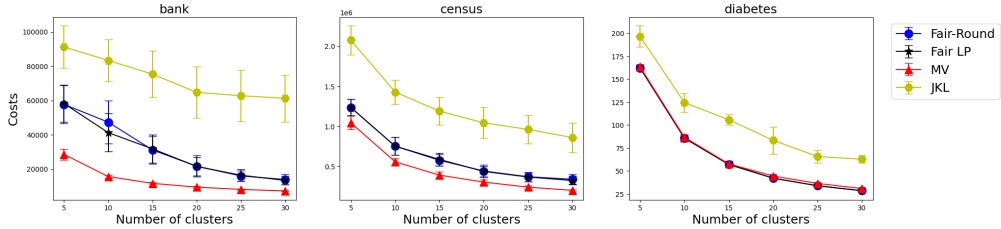

Figure 3: Comparison of $k$-MEANS clustering cost, between our algorithm Fair-Round, the algorithm in [25] (denoted as JKL), and the algorithm in [29] (denoted as MV), on average of 10 random samples of size 1000 each. We also plot the LP cost which is a lower bound on the optimum cost (denoted as Fair-LP.

**Runtime analysis.** We run our experiments in Python 3.8.2 on a MacBook Pro with 2.3 GHz 8-Core Intel Corei9 processor and 32 GB of DDR4 memory. We solve our linear programs using the Python API for CPLEX [22]. We demonstrate that even though solving an LP on the entire instance is time consuming, our sparsification step tremendously improves on the runtime (Figure 4) while increasing the clustering cost or fairness performance by a only a negligible margin (see Appendix A.4 for the cost and fairness results). Below, the blue line with circles shows Fair-Round with "vanilla" LP

solver, and the green line with upside-down triangles Sparse-Fair-Round is the runtimes with the sparse LP solution.

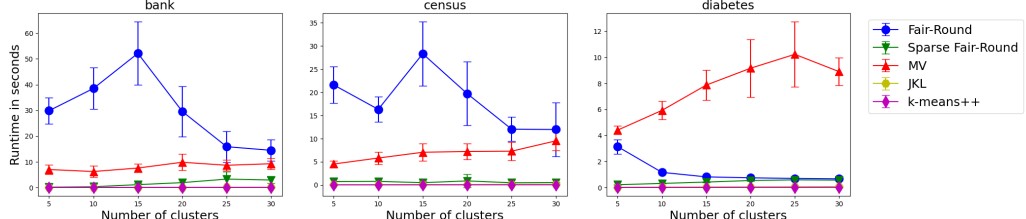

Figure 4: Comparison runtime with $k$-MEANS objective, between our algorithm Fair-Round, the algorithm in [25] (denoted as JKL), the algorithm in [29] (denoted as MV), and $k$-Means++ on average of 10 random samples of size 1000 each. Here, $\delta = 0.3$, $0.05$ and $0.01$ for bank, census, and diabetes. Note that [25], $k$-Means++, and our sparsified algorithm Sparse-Fair-Round have very small differences in their running times.

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
