# A    Complementary results for $k$-MEANS

In this section we provide experiments to further elaborate on our results from Section 4.

## A.1    Fairness histograms

Recall from Section 4.1 that the violation array for any solution $T \subseteq X$ is defined as $\theta_T$ over $v \in X$ as $\theta_T(v) = d(v, T)/r(v)$. Here we have the complete set of violation histograms similar to Figure 2. As evident, our algorithm is significantly fairer than [29] and closely matching [25].

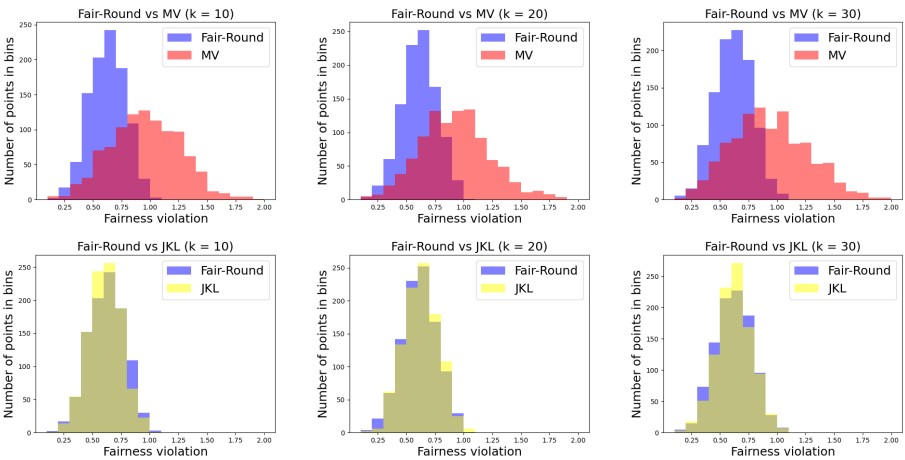

Figure 5: Histograms of violation vectors on `bank` with $k$-MEANS objective comparing Fair-Round with the algorithm in [25] (denoted as JKL) and the algorithm in [29] (denoted as MV), on average of 10 random samples of size 1000 each.

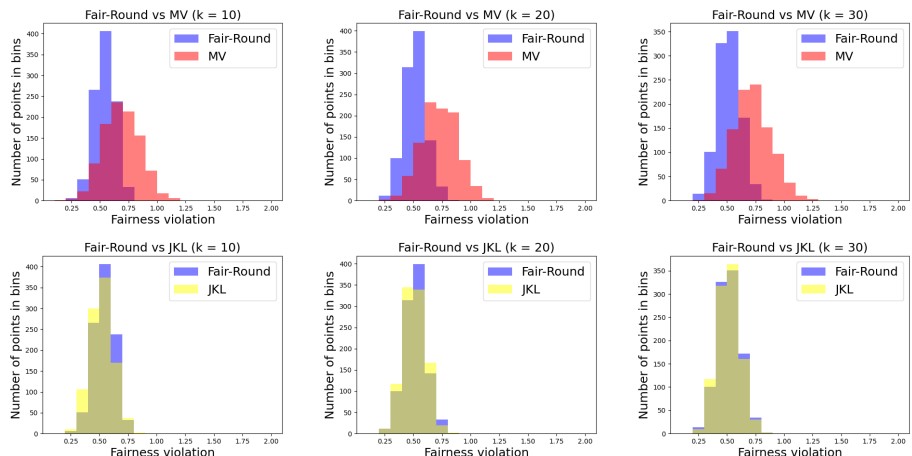

Figure 6: Histograms of violation vectors on `census` with $k$-MEANS objective comparing Fair-Round with the algorithm in [25] (denoted as JKL) and the algorithm in [29] (denoted as MV), on average of 10 random samples of size 1000 each.

## A.2    Relaxed-Fair-Round plots

As mentioned in Figure 2, allowing the same fairness radii violation as [29] makes our algorithm to give better cost and slightly better fairness violation than [29]. This is depicted in Figure 8 and Figure 9 respectively.

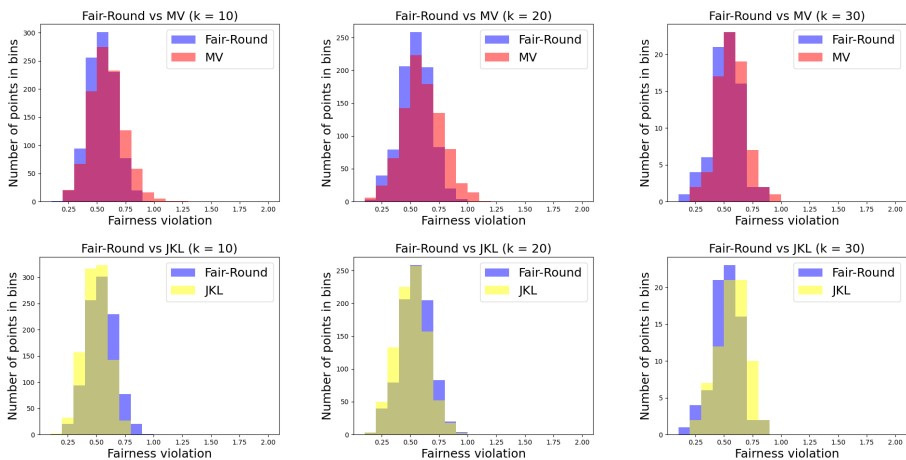

Figure 7: Histograms of violation vectors on `diabetes` with $k$-MEANS objective comparing Fair-Round with the algorithm in [25] (denoted as JKL) and the algorithm in [29] (denoted as MV), on average of 10 random samples of size 1000 each.

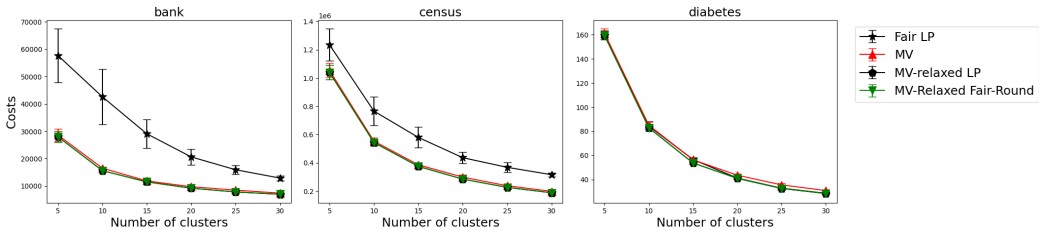

Figure 8: Comparison of $k$-MEANS objective cost, between [29] (denoted as MV), and our algorithm with radii relaxed as MV, Relaxed-Fair-Round, on average of 10 random samples of size 1000 each.

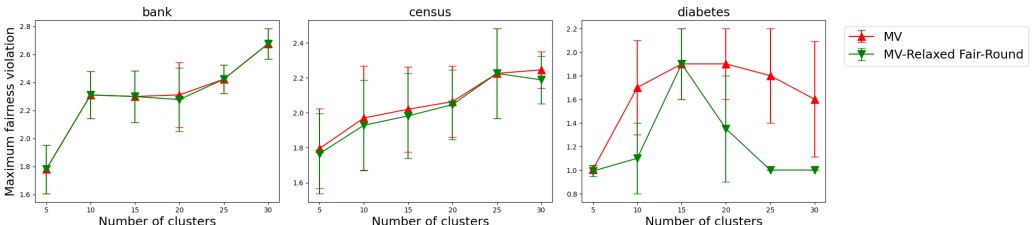

Figure 9: Comparison of maximum fairness violation with $k$-MEANS objective, between [29] (denoted as MV), fairLP cost, our algorithm with radii relaxed as MV, called Relaxed-Fair-Round, with the corresponding LP solution on average of 10 random samples of size 1000 each. Note, Relaxed-Fair-Round, MV, and the LP cost with relaxed radii match very closely.

## A.3    Cost of fairness

In this section, we demonstrate how the LP cost changes as we allow the points to violate the fairness radius by a varying constant factor. As previously mentioned in Section 4.2, the LP cost is used as a proxy for **opt**. The plots show what is called "the cost of fairness" for $k = 20$ across all the datasets. As we relax fairness constraints, the LP costs drops but the slope varies across datasets. The trend seems to be that in datasets where cost of fairness is not much affected by violation, the gap between our cost and [29] also seems to be lower (in Figure 3) as expected.

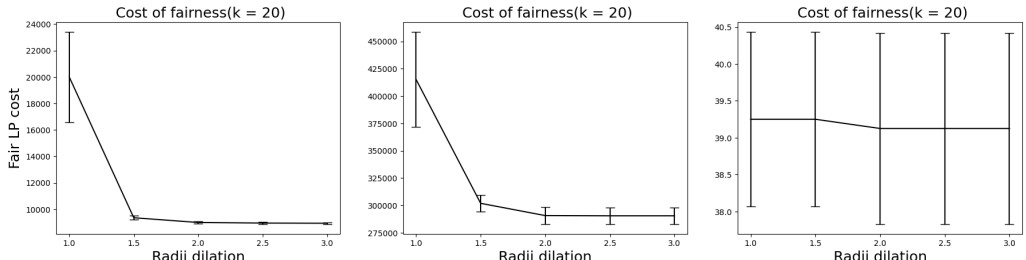

Figure 10: Comparison of LP cost for $k$-MEANS objective with varying constants dilation of radii, on average of 10 random samples of size 1000 each.

### A.4  **Sparse-Fair-Round** plots

As demonstrated in Figure 4, using Lemma 5 with $\delta = 0.3$, $0.05$ and $0.01$ for `bank`, `census`, and `diabetes` considerably decreases the LP solving time, hence, the overall runtime of our algorithm. Here we show that fairness and clustering cost are only slightly affected in Figure 11 and Figure 12 respectively.

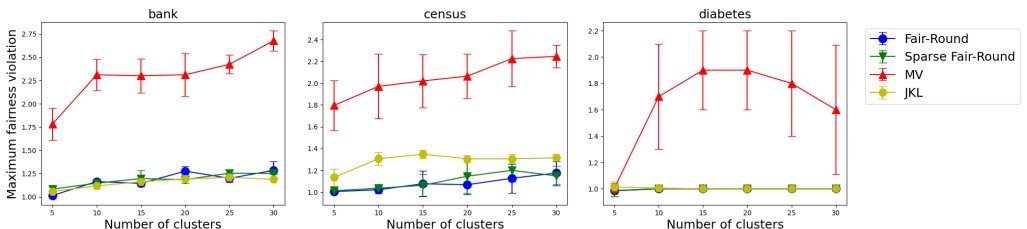

Figure 11: Comparison of maximum fairness violation with $k$-MEANS objective, between our algorithm before and after sparsification, the algorithm in [25] (denoted as JKL), and the algorithm in [29] (denoted as MV), on average of 10 random samples of size 1000 each.

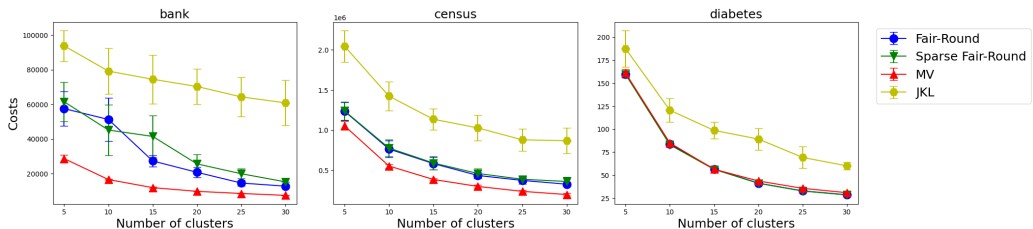

Figure 12: Comparison of $k$-MEANS clustering cost, between our algorithm before and after sparsification, the algorithm in [25] (denoted as JKL), and the algorithm in [29] (denoted as MV), on average of 10 random samples of size 1000 each.

## B  Results for $k$-MEDIAN

We repeat the experiments from Section 4 for $k$-MEDIAN objective and observe the same trends. We start off by plotting maximum fairness violations analogous to Figure 1.

Next, we compare the cost of Fair-Round with the algorithm in [25] (denoted as JKL) and the algorithm in [29] (denoted as MV). The results are similar to Figure 3 for $k$-MEANS.

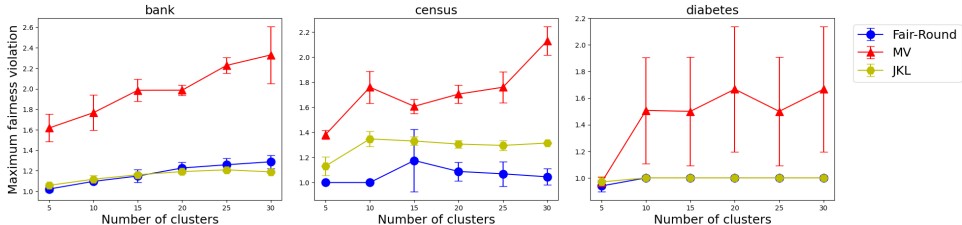

Figure 13: Comparison of maximum fairness violation with $k$-MEDIAN objective, between our algorithm Fair-Round, the algorithm in [25] (denoted as JKL), and the algorithm in [29] (denoted as MV), on average of 10 random samples of size 1000 each.

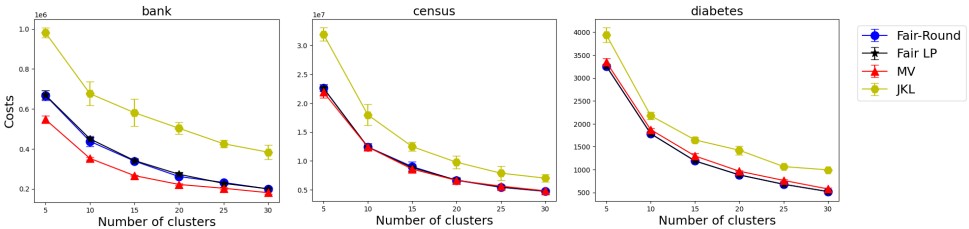

Figure 14: Comparison of $k$-MEDIAN clustering cost, between our algorithm Fair-Round, JKL, MV , on average of 10 random samples of size 1000 each. We also plot the LP cost which is a lower bound on the optimum cost (denoted as Fair-LP).

As for the runtime of our algorithm after sparsification, we run the same analysis as in Figure 4 but with the $k$-MEDIAN objective.

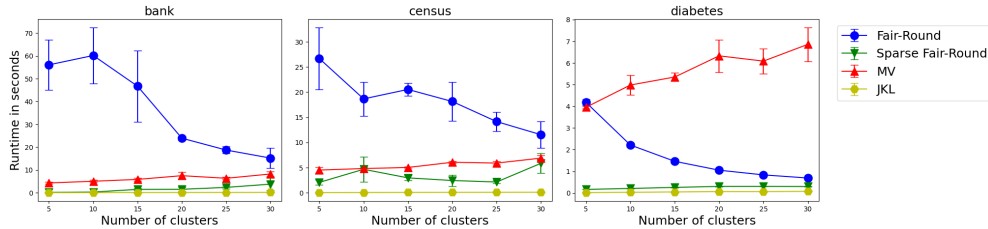

Figure 15: Comparison runtime with $k$-MEANS objective, between our algorithm Fair-Round before and after sparsification, the algorithm in [25] (denoted as JKL) and the algorithm in [29] (denoted as MV), on average of 10 random samples of size 1000 each. Here, $\delta = 0.3$, 0.05 and 0.01 for `bank`, `census`, and `diabetes`.