# OpenReview forum: "Better Algorithms for Individually Fair $k$-Clustering"
_NeurIPS.cc/2021/Conference — NeurIPS 2021 Poster_

### Official Review · Reviewer_RN5f · 2021-06-24

**Rating:** 7
**Confidence:** 5

**Summary:**

The authors of this paper consider the individual fair k-clustering problem, as introduced by Jung et al. [FORC 19]. In this model, for every point $j$ there is a given radius $r_j$, and the goal is to choose a set of centers such that each point has an open center within distance $r_j$ from it, and also one of the standard clustering objective functions is minimized.

Specifically, under the k-median and k-means objectives, the authors provide improved guarantees compared to the state of the art results (Mahabadi and Vakilian [ICML 2020]). For the k-median objective, they give an $(8,8)$-bicriteria algorithm, while the best known result was an $(84,7)$ one. For k-means their result is an $(4,8)$-bicriteria algorithm, while the result of Mahabadi and Vakilian was a $(\gamma,7)$ one, with $\gamma$ being a very large constant. (an $(a,b)$-bicriteria solution has objective cost at most $a$ times the optimal, and for every $j$ it has an open center within distance at most $b \cdot r_j$ from it).

The authors achieve their results via standard LP-rounding techniques.

**Limitations And Societal Impact:**

Yes

**Main Review:**

A COMMENT ON A PREVIOUSLY OVERLOOKED TRIVIAL SOLUTION:

Before I proceed with my actual review, I would like to make an observation regarding this particular fair clustering problem.

I remember after reading the Mahabadi and Vakillian paper, it was obvious to me that their local-search approach (although technically sound and very impressive) was unnecessarily complicated and far from efficient (in terms of the approximations achieved). Specifically, there is a trivial LP-based approach that can immediately give refined results. Focus on k-median for instance, and consider the standard LP for it with the additional constraint that $D_j = \sum_{i}x_{i,j} \cdot d(i,j) \leq r_j$, where $D_j$ is basically the fractional assignment of point $j$. Then any deterministic LP rounding approach that satisfies $\sum_{i}\hat{x}_{i,j} \cdot d(i,j) \leq \rho D_j$, would immediately yield results for the fair problem as well (here $\hat{x}$ is the integral assignment after the rounding). For instance, the rounding from "A Dependent LP-rounding Approach for the k-Median Problem" by Charikar and Li would give $\rho = 9$ and subsequently an $(9,9)$-bicriteria algorithm (I'm actually referring to the bundling process they use to solve matroid-median). In case I'm missing something (though I have discussed this with a couple fellow researchers), it's surprising how such a trivial solution was overlooked by Mahabadi and Vakillian.

CLARITY OF THE PAPER:

The paper is very easy to follow and the authors have done a great job with regards to writing. I really enjoyed reading this work.

CORRECTNESS:

I went though all the arguments as carefully as I could, and I believe that all the theoretical claims are sound.

SIGNIFICANCE - ORIGINALITY:

The main contribution of this work is improved results compared to the Mahabadi and Vakillian paper. However, I think that a more accurate comparison is with the trivial solution I mentioned earlier. Compared to that, the authors employ their own rounding approach, instead of just using another one as a black-box. This gives indeed an improvement, but under this comparison the refinement is not that substantial. Moreover, I don't think that the details of their technique are very novel, since similar rounding ideas have appeared before.

Overall, we have a problem that has a trivial solution which was overlooked by the previous "state-of-the-art" results. Compared to the trivial solution, the authors indeed provide refined results, yet 1) the improvement is not that substantial and 2) the techniques used are not really original. Nonetheless, it would be beneficial (to the research community) if the results of this paper are published, since they show a much more accurate algorithmic understanding of the problem (compared to the Mahabadi and Vakillian paper). My concern is that they might not be novel enough technique-wise to meet the high NeurIPS threshold.

MINOR COMMENT REGARDING THE EXPERIMENTS:

To test the scalability of the sparsification process it would have been better to run for a larger sample of points (1000 points seem already too few).

MOTIVATION:

When this model was first introduced I was skeptical about it, in the sense of believing that it doesn't actually capture any notion of fairness. It felt more like a robust version of k-median/k-means, where you want to give an additional covering guarantee to each client. From my perspective, Jung et al. did not do a good job justifying the fairness aspect of the problem (same thing with Mahabadi and Vakillian, although this was justified in the sense of them studying an already established model).

My issue stems from the fact that this notion of individual fairness diverges from the definition given by Dwork et al. in their seminal work "Fairness through Awareness" [ITCS 2012] (a solution is individually fair iff it treats similar points similarly).

I understand that the authors did not have to justify the model from a fairness perspective, since it is an already established one, and hence I reviewed their paper based only on its own merit without letting doubts of the model affect my judgement. However, because previous work lacks a convincing discussion on why this particular constraint captures fairness, it would be beneficial to the community if someone does this. Therefore, I recommend that the authors add a detailed discussion about this in their paper.



**Time Spent Reviewing:**

3 hours reading the paper + 1 hour thinking about the review + 1 hour writing the review

---

> ### Author Response · Authors · 2021-08-10
> **Response to reviewer RN5f**
>
> First, we thank the reviewer for their careful reading.
>
> We would like to point out that the "trivial solution" proposed by the reviewer may not work (unless we have misunderstood). The reviewer states that ``any deterministic LP rounding approach that atisfies $\sum_i \hat{x}_{ij}\cdot d(i,j) \leq \rho D_j$ would immediately yield results for the fair problem as well (here $\hat{x}$ is the integral assignment after rounding."
>
> This is absolutely correct, but the key word above is deterministic. Charikar-Li is, however, is randomized, and the LHS is in expectation. And even after applying concentration bounds, one cannot obtain such a statement for all clients. Indeed, such a rounding is \textbf{impossible}, and therefore many generalizations of k-median like ordered k-media a la Aouad-Segev (Math Prog 2021) and Byrka et al (STOC 2018), and minimum norm clustering (Chakrabarty-Swamy, STOC 2019) require new ideas.
>
> Here is the simple example : consider (k+1) points with every pair at distance 1. The LP can "cheat" and make every client pay 1/(k+1): simply open every point to k/k+1 fraction. The optimal solution must make some client pay 1, which (k+1) times more. Hopefully, this clarifies why MV had to do more work, and why our rounding also has to be a little more careful. We do, however, agree that the rounding techniques are not novel, and it connects ideas from different papers. However, one point of our paper is that they give better results than the existing papers and not only that, they can be made to run fast by using the sparsification idea in practice. Usually, LP solving is a no-go for large problems, but our sparsification idea leads to faster running time.
>
> The reviewer also mentions that we should do runtime analysis on a large sample of points. We would like to mention that as previous work did not scale well, we had to limit the experiments to 1000 points to be able to compare with the literature. We could add additional runtime analysis just to see how our algorithm performs by increasing the number of points, but we would not have any benchmarks to compare with. Still, we will definitely consider including this analysis in the next version of the paper.
>
> The reviewer also suggests that we add a discussion to motivate the notion of fairness. We also agree, and given more space, we will add this in the next version of our paper.

---

> > ### Comment · Reviewer_RN5f · 2021-08-23
> > **Response to rebuttal comments**
> >
> > Regarding the comment on the "trivial" solution, I agree that the rounding should be deterministic. I had already mentioned that in my initial review. However, the specific rounding from Charikar and Li that I was referring to is not the one that achieves the 3.25 ratio (given the randomized nature of it I agree that it is impossible to apply it for this problem). In the full-version of their paper, Charikar and Li provide a deterministic rounding for matroid-median, that achieves a 9 approximation ratio (this is an application of their bundling approach with some simple matching arguments). Since k-median is a special case of matroid-median, the aforementioned 9-approximation solves the former problem as well, while satisfying the $\sum_{i}\hat{x}_{i,j}d(i,j) \leq \rho D_j$ requirement deterministically. In my initial review I had explicitly mentioned that I was actually referring to that algorithm and not the 3.25 one.

---

> > > ### Author Response · Authors · 2021-08-25
> > > **Response to comment**
> > >
> > > Thank you for the comment, but we think we are still at odds. In the deterministic algorithm for the matroid median, we don't think Charikar-Li ever assert that the cost paid by client j, which they call $SOL_j$, is at most 9 times what the client pays in the LP. What they do say (we henceforth refer the full version https://cse.buffalo.edu/~shil/papers/KM325-Extended.pdf), in the proof of Theorem 3 in Appendix A on page 13, is that $SOL_j$  is at most $4LP_j + 5LP^{(2)}_j$. Here,  $LP^{(2)}_j$ is what client j pays in a basic feasible solution $y^{(2)}$ in a *new* LP, called LP(2) in their paper.  This $y^{(2)}$ is used in their proof of $SOL_j <= 4LP_j + 5LP^{(2)}_j$ in Lemma 7.
> > >
> > > However, they never claim $LP^{(2)}_j$ is at most $LP_j$ client-by-client. In Lemma 4, they claim the *sum* of $LP^{(2)}_j$'s is at most the sum of $LP^{(1)}_j$'s (which is at most the sum of $LP_j$)'s. This they show in a client-by-client way (which may have led to the confusion) as follows. One takes the original y-solution of $LP^{(1)}$, and shows (i) y is a *feasible* solution for $LP(2)$, and (ii) y's contribution to $LP(2)$ is at most y's contribution to $LP(1)$, client-by-client. And thus the optimum *value* of $LP(2)$ is at most that of $LP(1)$. However, this y may *not* be a basic feasible solution of $LP(2)$, and basic feasibility is crucial for their rounding in Lemma 7. And when one moves to a basic feasible solution, this per-client guarantee is destroyed.
> > >
> > > Indeed, as we proposed in our previous rebuttal, such a "per-client" guarantee is not possible **for any algorithm** for k-median. And in that example, one would find that no matter how you choose the solution for $LP^{(2)}$, some client j will have $LP^{(2)}_j \gg LP_j$.
> > >
> > > In general, providing the fairness guarantee when you would redirect clients of a bundle to its partner in the matching is not trivial. In our case, we could only make the argument work by changing the definition of a bundle similar to Alamdari and Shmoys. When creating the bundles, looking at the LP cost of a point v is not enough. Turns out, defining the bundles based on the *min* of r(v) and a factor of its LP cost, is the right way to go. This is a simple yet non-trivial modification especially that when the $r(v)$’s differ across points it is not immediately clear if this new definition can help us ensure fairness.
> > >
> > > Please let us know if we are still missing something.

---

> > > > ### Comment · Reviewer_RN5f · 2021-08-25
> > > > **Response to comment**
> > > >
> > > > You are actually right. The confusion resulted by how Lemma 4 is proven in Charikar and Li, where there is a client-by-client comparison of how much $y^{(1)}$ contributes to both LPs. However, I now see that this does not imply $LP^{(2)}_j \leq LP^{(1)}_j$. Given this realization which means that there is no trivial solution to the problem, I'm increasing my score.

---

### Official Review · Reviewer_mdjA · 2021-07-16

**Rating:** 7
**Confidence:** 3

**Summary:**

This submission proposes an approximation algorithm for individual fair k-clustering in a metric space. Here, individual fairness means that each point v has an individual radius $r(v)$ and we require an open facility within radius $r(v)$ of $v$. We may use the radius $r(v)$ to provide each point $v$ with a "fair share" of its assigned facility by, for instance, choosing $r(v)$ as the smallest radius such that $n/k$ points lie within radius $r(v)$ of $v$; as originally proposed by Jung, Kannan, and Lutz.

The approximation algorithm is the main contribution of the submission. It works for any p-norm objective and is bi-criterial, guaranteeing an $2^{1+2/p}$-approximation of the  objective while ensuring an open facility within radius $8r(v)$ of each point $v$. However, it requires a metric space which means, in particular, that all facilities/centers have to be chosen from the point set.

The algorithm roughly works in the following way: First, the authors compute an optimum solution to a linear programming relaxation of the problem. The solution is turned into an input for a filter function by Plesnik/Rita, Moro and Cortez; this function replaces the original point set by a smaller set of representatives. In the original version of the filter, the function may be run until only $k$ representatives remain (a point may represent another if it can serve as the point's facility; hence, in the original filter, the representatives are then chosen as the facilities to open). In this case, however, it may be that the filter stops early as the fairness restriction on the connection radius may make it impossible to cover all points with $k$ representatives. Hence, a final step is required to reduce the set of representatives down to $k$ points. This is done via an LP-rounding algorithm and turns out to be a substantial amount of work. Finally, in order to improve the running time of the algorithm, the authors propose a sparsification technique for the LP-relaxation (whose size is otherwise quadratic in the number of points and could be prohibitive for larger point sets).

An empirical part shows improvements over the previous approach.

**Limitations And Societal Impact:**

See above; it would like the submission to be more explicit about having to choose the centers/facilities from the input points.

**Main Review:**

The paper is exceptionally well written and I found it good to follow even through the substantial technical parts. The authors state that their work is mostly a combination of existing ideas. I feel this is a rather modest description and found the work to be quite non-incremental. I particularly like how the existing approach is extended to the individual fairness model in a non-trivial way. The experimental part is extensive; certainly larger than I would have expected for a paper with a sizeable theoretical contribution. The sparsification shows that clearly practicability was a priority for the authors.

As the weaker sides of the submission, I only see the assumption that all facilities must be chosen from the input points; the  approximation results for k-means (l2-objective, in language of the paper) from the literature generally assume a Euclidean space and permit the use of non-input points as facilities. The cited reference [1] considers Euclidean k-median. The assumption may be removed, but only at the loss of a factor of 2 in the approximation guarantee.

This is not so much of a limitation, but the distinction could be more explicit in the paper. In particular, having a decent LP-rounding and sparsification for Euclidean k-means without having to invest a factor of 2 would be a big result.

**Time Spent Reviewing:**

3

---

> ### Author Response · Authors · 2021-08-10
> **Response to reviewer mdjA**
>
> We thank the reviewer for their encouraging remarks and are happy that they had a good time reading our paper. To address the reviewer's remark on clarity, we will state the following fact explicitly to avoid any confusion: that we choose the centers among the points, and "The assumption may be removed, but only at the loss of a factor of 2 in the approximation guarantee." Moreover, based on our experience we believe that the factor of 2 loss is pessimistic in real-world applications as often times there are dataset points very close to optimal centers.
>
> We agree with the reviewer that "In particular, having a decent LP-rounding and sparsification for Euclidean k-means without having to invest a factor of 2 would be a big result." Indeed if we find such method it would be a great result of its own, independent from this paper.

---

### Official Review · Reviewer_XHqy · 2021-08-01

**Rating:** 5
**Confidence:** 3

**Summary:**

The paper studies the problem of \ell_p norm and center-based clustering with a
fairness consistent. The fairness constraint roughly says that no point should
be further from its assigned center than it is from its (n/k)-th furthest
neighbor. The authors consider a Linear Programming based (approximation)
algorithm for this problem.


**Ethical Concerns:**

There are none.

**Limitations And Societal Impact:**

See above for limitations.

Regarding societal impact, while I think there is inadequate discussion about it in the paper, I don't hold this against the authors. It's not clear what is to be expected from a paper like this.


**Main Review:**

Center based clustering is widely studied problem where the basic problem of
clustering, i.e. partitioning data is framed as a combinatorial optimization
problem of picking k centers such that some combination of the distance of the
points from their nearest center (typically the sum -- though can be max in the
case of k-center) is minimized. These problems are well-studied -- most of them
are NP-hard, and in several cases approximation algorithms to some factors are
known. There are also hardness of approximation results known for several
settings -- though some gaps in our understanding remain. Since the main
purpose of the paper is not to discuss these problems per se, this should
suffice for the purpose of this review.

The goal in this paper is to have an additional "fairness" constraint -- where
for each point v in the dataset, there is a quantity r(v) and the requirement
(hard constraint) is that the center assigned to v should not be at a distance
of more than r(v) from this point. Of course, in some cases this may be
impossible and relaxations can be considered. There is also the standard k-*
objective, where * can be median, mean, center etc. Let me call these
objectives A (radius r(v)) and B cost of the k-* value. Then an (\alpha, \beta)
bi-criteria approximation algorithm guarantees that the distance of v from its
assigned center is no more than \alpha r(v) and also that the overall cost of
the objective B for the k-* value is no more than \beta OPT.

The authors design LP-based algorithms that can achieve "better" bi-criteria optimization algorithms that were available in prior work.

The main shortcoming I see in this work is that this particular criterion has been proposed very recently, as recent as 2020, and in particular the authors don't explain very well why this is an interesting objective to consider. In fairness, the authors do say that their focus is on the "algorithmic" rather than the fairness part of the work, but it is not clear to me that this is necessarily an interesting algorithmic problem to solve if the fairness objective is not well-motivated enough, and the authors do little to motivate it. Furthermore, the actual algorithmic techniques are not particularly novel or impressive, though it does improve on the current state of affairs. Thus, I am relatively lukewarm about this paper.


**Time Spent Reviewing:**

3.5

---

> ### Author Response · Authors · 2021-08-10
> **Response to reviewer XHqy**
>
> Firstly, we thank the reviewer for their critique. The main concern seems to be the notion of fairness. As the reviewer points out, this notion of fairness is not due to us, and perhaps this (and the usual space constraints) explains why we did not go into explaining why the model is relevant.
>
> Yes, this notion of fairness is new, but as the reviewer surely knows, this area is fast evolving. We do feel this notion of fairness (due to Jung, Kannans and Lutz) makes sense in many application domains. If we think of cluster centers as, say, hospitals, any notion of "individual fairness" must be a function of the distances each client travels to the nearest open center. There is the common tension between individual good and the social good, and the way Jung et al address this, is by stating some "levels of service" for every point depending on how many other points are in its neighborhood. The notion is arguable, but we believe the notion is natural. In our paper, we actually provide a "knob" for "levels of service" for different clients (which can be set by the application domain expert) although we restrict our experiments to the Jung et al model.
>
> The other point the reviewer makes is that this question is not necessarily an interesting algorithmic question if one does not "buy" the fairness definition. Here, we would have to disagree. We think that this constrained clustering question is interesting in its own right. After all, as the reviewer knows, the objectives to a clustering problem is just a proxy, and it is a natural question : which objective functions are tractable? There is some recent work on this by (Byrka et al STOC 2018, Aouad-Segev Math Programming 2020, Chakrabarty-Swamy STOC 2019, etc) which look at general symmetric norms on the cost vector. The question the fairness problem leads us to, is actually a *special* case of the asymmetric norm case (we do not make any of these connections in the paper explicit mainly due to space reasons, but here we are talking to the expert reviewer). And so, the algorithmic question is quite interesting. Having said that, we do agree that the algorithmic techniques we use for rounding the LP is "not particularly novel"; but we wanted to show that LP-rounding algorithms can actually solve decently large problems even in practice after using the sparsification step (which uses the Jung et al model) and we felt that this is something the NeurIPS and the fairness in clustering community would care about.

---

### Official Review · Reviewer_UP3J · 2021-08-02

**Rating:** 6
**Confidence:** 3

**Summary:**

This paper introduces a new algorithm for fair k-clustering problem(specifically when cost is measure with l_p norm for different p values) based on LP rounding. It provides a bi-criteria approximation guarantee for the clustering objective and experiments are provided to validate the theoretical properties. It is also shown how the runtime of the algorithm can be improved using sparsification with minimal effects on the objective.

**Ethical Concerns:**

None.

**Limitations And Societal Impact:**

Some limitations are discussed regarding the theoretical analysis of violation vectors.
It may be worthwhile comparing this work with other notions of fair clustering, for example when the objective is maintaining similar cost for every cluster (see "Fair Clustering via Equitable Group Representations" by Abbassi, Bhaskara, and Venkatasubramanian).


**Main Review:**

# Originality

Although LP formulations have been studied for various clustering settings under different constraints, this paper introduces a nice formulation that handles fairness constraints and I believe that the rounding procedure and the analysis contain a decent amount of original work.

# Quality

The quality of the writing and the presentation are good. This result has been placed among the relevant prior work well. The experiment complements the theoretical guarantees provided.

# Clarity

The clarity of the paper for the most part is good. However, I believe the following places need more clarification.
- In the experiments, when looking at the violations, what is considered the ground truth clustering? Do the datasets have cluster labels and if so do they adhere to the k-means(or k-median) optimal clustering? or output from some standard algorithm(eg: k-mean++) is considered the ground truth?
- It is not clear why Fact 4. in line 186 hold. Don't the updates in Algorithm 2, line 13 or line 19 affect that condition on y?

# SIgnificance

Fair k-clustering is an important problem for the NeurIPS community in my opinion and the improvements on the running time and the approximation guarantees simultaneously for the clustering objective and fairness (where prior works are lacking) carries a substantial significance.

Overall, I believe this paper adds decent contributions.


**Time Spent Reviewing:**

6

---

> ### Author Response · Authors · 2021-08-10
> **Response to reviewer UP3J**
>
> We thank the reviewer for their insightful review. Let us start with the clarification points.
>
> The first question is what we use as our ground-truth labels. Our datasets (which are used in previous work) are all unlabeled and our ground-truth is the optimal clustering: the clustering with the minimum objective value, or clustering loss in other terms. Since computing the actual optimal solution is computationally infeasible, we compute a lower-bound on the cost of the optimal solution via solving a linear program (LP). Since we do favorably with respect to this bound, we are guaranteed to be at least as good when compared to the optimal fair solution. Also since the output of standard algorithms (like k-means++) can be shown to be extremely unfair, we cannot use them as ground truth.
>
> As for the second question on why Fact 4 holds, since the given $y$ is promised to be from an LP solution, the total $y$ mass on the points equals k by the LP constraints in line 127. Then, you will see that the modifications on the $y$ vector we make before line 11 of the algorithm, is to move the y mass from all the points to only on members in S without increasing or decreasing the total y mass. Similarly, lines 13 and 19 are just taking $\delta$ mass from one member of $S$ and adding it to another member in $S$ so the total mass stays the same.
>
> We appreciate it that the reviewer brought these to our attention and will address these clarity issues in the next version of our paper.
>
> To address the comments on "Limitations And Societal Impact" we agree that it will be interesting to do cost vector analysis for existing algorithms on other notions of fairness as well. The only issue would be that the other papers, for example "Fair Clustering via Equitable Group Representation" are optimizing for a different notion of fairness altogether and it might not be an apples-to-apples comparison to compare their cost vectors with ours. Indeed, when it comes to using any of these algorithms in production, it is very important to do a thorough analysis of the cost vector and decide if it makes sense to use them in real world. We are happy that we give an example of such analysis and establish its importance. But we also think that running this analysis on different notions of fairness, although very interesting and important, is beyond the scope of this paper.

---

> > ### Comment · Reviewer_UP3J · 2021-08-27
> > **Thank you for clarifications.**
> >
> > I thank the authors for the clarifications for my concerns. I read the other reviews and author responses. I agree with the authors that an exhaustive comparison with other notions of fairness should not be required however, I believe that as a part of the motivation(as other reviewers have brought up), authors can place the fairness objective used in this paper among other notions of fairness in the discussion which is a minor concern for the revised version.
> > I maintain my score for the acceptance of the paper.

---

> > > ### Author Response · Authors · 2021-08-28
> > > **Response to Comments**
> > >
> > > We would like to thank the reviewer for their comment. We also agree that adding a discussion comparing with other relevant fairness notions is prudent for the motivation discussion. We will add this in the next version of our paper.

---

### Decision · Program_Chairs · 2021-09-27

**Decision:**

Accept (Poster)

**Comment:**

The authors consider the problem of \ell-p objective (e.g. k-median, k-means) centre based clustering with fairness constraints. The notion of fairness is that no point v should be at a distance of more than r(v) from its respective centre (where there are some conditions on r(v) to ensure feasibility). This notion of fairness was introduced in prior work and the authors main contribution is algorithmic. The algorithmic techniques while standard do improve on the (very recent) state of the art; thus the paper could be of interest to the NeurIPS community. There was a lot of discussion about the notion of fairness as well. It is suggested that the authors critique the notion (from all angles) more effectively, given how new this notion is, in order for the paper to have the most influence.